# ELEPHANT NEURAL NETWORKS: BORN TO BE A CONTINUAL LEARNER

## ABSTRACT

Catastrophic forgetting remains a significant challenge to continual learning for decades. While recent works have proposed effective methods to mitigate this problem, they mainly focus on the algorithmic side. Meanwhile, we do not fully understand what architectural properties of neural networks lead to catastrophic forgetting. This study aims to fill this gap by studying the role of activation functions in the training dynamics of neural networks and their impact on catastrophic forgetting. Our study reveals that, besides sparse representations, the gradient sparsity of activation functions also plays an important role in reducing forgetting. Based on this insight, we propose a new class of activation functions, elephant activation functions, that can generate both sparse representations and sparse gradients. We show that by simply replacing classical activation functions with elephant activation functions, we can significantly improve the resilience of neural networks to catastrophic forgetting. Our method has broad applicability and benefits for continual learning in regression, class incremental learning, and reinforcement learning tasks. Specifically, we achieves excellent performance on Split MNIST dataset in just one single pass, without using replay buffer, task boundary information, or pre-training.

## 1 INTRODUCTION

One of the biggest challenges to achieving continual learning is the decades-old issue of *catastrophic forgetting* (French 1999). Catastrophic forgetting stands for the phenomenon that artificial neural networks tend to forget prior knowledge drastically when learned with stochastic gradient descent algorithms on non-independent and identically distributed (non-iid) data. In recent years, researchers have made significant progress in mitigating catastrophic forgetting and proposed many effective methods, such as replay methods (Mendez et al. 2022), regularization-based methods (Riemer et al. 2018), parameter-isolation methods (Mendez et al. 2020), and optimization-based methods (Farajtabar et al. 2020). However, instead of alleviating forgetting by designing neural networks with specific properties, most of these methods circumvent the problem by focusing on the algorithmic side. *There is still a lack of full understanding of what properties of neural networks lead to catastrophic forgetting.* Recently, Mirzadeh et al. (2022a) found that the width of a neural network significantly affects forgetting and they provided explanations from the perspectives of gradient orthogonality, gradient sparsity, and lazy training regime. Furthermore, Mirzadeh et al. (2022b) studied the forgetting problem on large-scale benchmarks with various neural network architectures. They demonstrated that architectures can play a role that is as important as algorithms in continual learning.

*The interactions between continual learning and neural network architectures remain to be under-explored.* In this work, we aim to better understand and reduce forgetting by studying the impact of various architectural choices of neural networks on continual learning. Specifically, we focus on activation functions, one of the most important elements in neural networks. Experimentally, we study the effect of various activation functions (e.g., Tanh, ReLU, and Sigmoid) on catastrophic forgetting under the setting of continual supervised learning. Theoretically, we investigate the role of activation functions in the training dynamics of neural networks. Our analysis suggests that not only sparse representations but also sparse gradients are essential for continual learning. Based on this discovery, we design a new class of activation functions, the elephant activation functions. Unlike classical activation functions, elephant activation functions are able to generate both sparse func-

tion values and sparse gradients that make neural networks more resilient to catastrophic forgetting. Correspondingly, the neural networks incorporated with elephant activation functions are named *elephant neural networks (ENNs)*. In streaming learning for regression, while many classical activation functions fail to even approximate a simple sine function, ENNs succeed by applying elephant activation functions. In class incremental learning, we show that ENNs achieve excellent performance on Split MNIST dataset in one single pass, without using any replay buffer, task boundary information, or pre-training. In reinforcement learning, ENNs improve the learning performance of agents under extreme memory constraints by alleviating the forgetting issue.

## 2 INVESTIGATING CATASTROPHIC FORGETTING VIA TRAINING DYNAMICS

Firstly, we look into the forgetting issue via the training dynamics of neural networks. Consider a simple regression task. Let a scalar function $f_{\boldsymbol{w}}(\boldsymbol{x})$ be represented as a neural network, parameterized by $\boldsymbol{w}$, with input $\boldsymbol{x}$. $F(\boldsymbol{x})$ is the true function and the loss function is $L(f, F, \boldsymbol{x})$. For example, for squared error, we have $L(f, F, \boldsymbol{x}) = (f_{\boldsymbol{w}}(\boldsymbol{x}) - F(\boldsymbol{x}))^2$. At each time step $t$, a new sample $\{\boldsymbol{x}_t, F(\boldsymbol{x}_t)\}$ arrives. Given this new sample, to minimize the loss function $L(f, F, \boldsymbol{x}_t)$, we update the weight vector by $\boldsymbol{w}' = \boldsymbol{w} + \Delta_{\boldsymbol{w}}$ where $\Delta_{\boldsymbol{w}}$ is the weight difference. With the stochastic gradient descent (SGD) algorithm, we have $\boldsymbol{w}' = \boldsymbol{w} - \alpha \nabla_{\boldsymbol{w}} L(f, F, \boldsymbol{x}_t)$, where $\alpha$ is the learning rate. So $\Delta_{\boldsymbol{w}} = \boldsymbol{w}' - \boldsymbol{w} = -\alpha \nabla_{\boldsymbol{w}} L(f, F, \boldsymbol{x}_t) = -\alpha \nabla_f L(f, F, \boldsymbol{x}_t) \nabla_{\boldsymbol{w}} f_{\boldsymbol{w}}(\boldsymbol{x}_t)$. By Taylor expansion,

$$f_{\boldsymbol{w}'}(\boldsymbol{x}) - f_{\boldsymbol{w}}(\boldsymbol{x}) = -\alpha \nabla_f L(f, F, \boldsymbol{x}_t) \langle \nabla_{\boldsymbol{w}} f_{\boldsymbol{w}}(\boldsymbol{x}), \nabla_{\boldsymbol{w}} f_{\boldsymbol{w}}(\boldsymbol{x}_t) \rangle + O(\Delta_{\boldsymbol{w}}^2), \tag{1}$$

where $\langle \cdot, \cdot \rangle$ denotes Frobenius inner product or dot product depending on the context. In this equation, since $-\alpha \nabla_f L(f, F, \boldsymbol{x}_t)$ is unrelated to $\boldsymbol{x}$, we only consider the quantity $\langle \nabla_{\boldsymbol{w}} f_{\boldsymbol{w}}(\boldsymbol{x}), \nabla_{\boldsymbol{w}} f_{\boldsymbol{w}}(\boldsymbol{x}_t) \rangle$, which is known as the neural tangent kernel (NTK) (Jacot et al. 2018). Without loss of generality, assume that the original prediction $f_{\boldsymbol{w}}(\boldsymbol{x}_t)$ is wrong, i.e. $f_{\boldsymbol{w}}(\boldsymbol{x}_t) \neq F(\boldsymbol{x}_t)$ and $\nabla_f L(f, F, \boldsymbol{x}_t) \neq 0$. To correct the wrong prediction while avoiding forgetting, we expect this NTK to satisfy two properties that are essential for continual learning:

**Property 2.1** (error correction). *For $\boldsymbol{x} = \boldsymbol{x}_t$, $\langle \nabla_{\boldsymbol{w}} f_{\boldsymbol{w}}(\boldsymbol{x}), \nabla_{\boldsymbol{w}} f_{\boldsymbol{w}}(\boldsymbol{x}_t) \rangle \neq 0$.*

**Property 2.2** (zero forgetting). *For $\boldsymbol{x} \neq \boldsymbol{x}_t$, $\langle \nabla_{\boldsymbol{w}} f_{\boldsymbol{w}}(\boldsymbol{x}), \nabla_{\boldsymbol{w}} f_{\boldsymbol{w}}(\boldsymbol{x}_t) \rangle = 0$.*

In particular, Property 2.1 allows for error correction by optimizing $f_{\boldsymbol{w}'}(\boldsymbol{x}_t)$ towards the true value $F(\boldsymbol{x}_t)$, so that we can learn new knowledge (i.e., update the learned function). If $\langle \nabla_{\boldsymbol{w}} f(\boldsymbol{x}), \nabla_{\boldsymbol{w}} f_{\boldsymbol{w}}(\boldsymbol{x}_t) \rangle = 0$, we then have $f_{\boldsymbol{w}'}(\boldsymbol{x}) - f_{\boldsymbol{w}}(\boldsymbol{x}) \approx 0$, failing to correct the wrong prediction at $\boldsymbol{x} = \boldsymbol{x}_t$. Essentially, Property 2.1 requires the gradient norm to be non-zero. On the other hand, Property 2.2 is much harder to be satisfied, especially for non-linear approximations. Essentially, to make this property hold, except for $\boldsymbol{x} = \boldsymbol{x}_t$, the neural network $f$ is required to achieve zero knowledge forgetting after one step optimization, i.e. $\forall \boldsymbol{x} \neq \boldsymbol{x}_t, f_{\boldsymbol{w}'}(\boldsymbol{x}) = f_{\boldsymbol{w}}(\boldsymbol{x})$. *It is the violation of Property 2.2 that leads to the forgetting issue.* For tabular cases (e.g., $\boldsymbol{x}$ is a one-hot vector and $f_{\boldsymbol{w}}(\boldsymbol{x})$ is a linear function), this property may hold by sacrificing the generalization ability of deep neural networks. In order to benefit from generalization, we propose Property 2.3 by relaxing Property 2.2.

**Property 2.3** (local elasticity). *$\langle \nabla_{\boldsymbol{w}} f_{\boldsymbol{w}}(\boldsymbol{x}), \nabla_{\boldsymbol{w}} f_{\boldsymbol{w}}(\boldsymbol{x}_t) \rangle \approx 0$ for $\boldsymbol{x}$ that is dissimilar to $\boldsymbol{x}_t$ in a certain sense.*

Property 2.3 is known as local elasticity (He & Su 2020). A function $f$ is locally elastic if $f_{\boldsymbol{w}}(\boldsymbol{x})$ is not significantly changed, after the function is updated at $\boldsymbol{x}_t$ that is dissimilar to $\boldsymbol{x}$ in a certain sense. For example, we can characterize the dissimilarity with the 2-norm distance. Although He & Su (2020) show that neural networks with nonlinear activation functions are locally elastic in general, there is a lack of theoretical understanding about the connection of neural network architectures and the degrees of local elasticity. In our experiments, we find that the degrees of local elasticity of classical neural networks are not enough to address the forgetting issue, as we will show next.

## 3 UNDERSTANDING THE SUCCESS AND FAILURE OF SPARSE REPRESENTATIONS

The above properties can help to deepen our understanding of the success and failure of sparse representations in continual learning. To be specific, we argue that sparse representations are effective

to reduce forgetting in linear function approximations, but are less useful in non-linear function approximations.

It is well-known that deep neural networks can automatically generate effective representations (a.k.a. features) to extract key properties from input data. The ability to learn useful features helps deep learning methods achieve great success in many areas (LeCun et al. 2015). In particular, we call a set of representations sparse when only a small part of representations is non-zero for a given input. Sparse representations are shown to help reduce the forgetting problem and the interference issues in both continual supervised learning and reinforcement learning (Shen et al. 2021, Liu et al. 2019). Formally, let $x$ be an input and $\phi$ be an encoder that transforms the input $x$ into its representation $\phi(x)$. The representation $\phi(x)$ is sparse when most of its elements are zeros.

First, consider the case of linear approximations. A linear function is defined as $f_{\boldsymbol{w}}(\boldsymbol{x}) = \boldsymbol{w}^\top \phi(\boldsymbol{x})$ : $\mathbb{R}^n \mapsto \mathbb{R}$, where $\boldsymbol{x} \in \mathbb{R}^n$ is an input, $\phi : \mathbb{R}^n \mapsto \mathbb{R}^m$ is a fixed encoder, and $\boldsymbol{w} \in \mathbb{R}^m$ is a weight vector. Assume that the representation $\phi(\boldsymbol{x})$ is sparse and non-zero (i.e., $\|\phi(\boldsymbol{x})\|_2 > 0$) for $\boldsymbol{x} \in \mathbb{R}^n$. Next, we show that both Property 2.1 and Property 2.3 are satisfied in this case. Easy to know that $\nabla_{\boldsymbol{w}} f_{\boldsymbol{w}}(\boldsymbol{x}) = \phi(\boldsymbol{x})$. Together with Equation (1), we have

$$f_{\boldsymbol{w}'}(\boldsymbol{x}) - f_{\boldsymbol{w}}(\boldsymbol{x}) = -\alpha \nabla_f L(f, F, \boldsymbol{x}_t) \phi(\boldsymbol{x})^\top \phi(\boldsymbol{x}_t). \tag{2}$$

By assumption, $f_{\boldsymbol{w}}(\boldsymbol{x}_t) \neq F(\boldsymbol{x}_t)$ and $\nabla_f L(f, F, \boldsymbol{x}_t) \neq 0$. Then Property 2.1 holds since $f_{\boldsymbol{w}'}(\boldsymbol{x}_t) - f_{\boldsymbol{w}}(\boldsymbol{x}_t) = -\alpha \nabla_f L(f, F, \boldsymbol{x}_t) \|\phi(\boldsymbol{x}_t)\|_2^2 \neq 0$. Moreover, when $\boldsymbol{x} \neq \boldsymbol{x}_t$, it is very likely that $\langle \phi(\boldsymbol{x}), \phi(\boldsymbol{x}_t) \rangle \approx 0$ due to the sparsity of $\phi(\boldsymbol{x})$ and $\phi(\boldsymbol{x}_t)$. Thus, $f_{\boldsymbol{w}'}(\boldsymbol{x}) - f_{\boldsymbol{w}}(\boldsymbol{x}) = -\alpha \nabla_f L(f, F, \boldsymbol{x}_t) \langle \phi(\boldsymbol{x}), \phi(\boldsymbol{x}_t) \rangle \approx 0$ and Property 2.3 holds. We conclude that by satisfying Property 2.1 and Property 2.3, sparse representations successfully help mitigate catastrophic forgetting in linear approximations.

However, for non-linear approximations, sparse representations can no longer guarantee Property 2.3. Consider an MLP with one hidden layer $f_{\boldsymbol{w}}(\boldsymbol{x}) = \boldsymbol{u}^\top \sigma(\mathbf{V}\boldsymbol{x} + \boldsymbol{b}) : \mathbb{R}^n \mapsto \mathbb{R}$, where $\sigma$ is a non-linear activation function, $\boldsymbol{x} \in \mathbb{R}^n$, $\boldsymbol{u} \in \mathbb{R}^m$, $\mathbf{V} \in \mathbb{R}^{m \times n}$, $\boldsymbol{b} \in \mathbb{R}^m$, and $\boldsymbol{w} = \{\boldsymbol{u}, \mathbf{V}, \boldsymbol{b}\}$. We compute the NTK in this case, resulting in the following lemma.

**Lemma 3.1** (NTK in non-linear approximations). *Given a non-linear function $f_{\boldsymbol{w}}(\boldsymbol{x}) = \boldsymbol{u}^\top \sigma(\mathbf{V}\boldsymbol{x} + \boldsymbol{b}) : \mathbb{R}^n \mapsto \mathbb{R}$, where $\sigma$ is a non-linear activation function, $\boldsymbol{x} \in \mathbb{R}^n$, $\boldsymbol{u} \in \mathbb{R}^m$, $\mathbf{V} \in \mathbb{R}^{m \times n}$, $\boldsymbol{b} \in \mathbb{R}^m$, and $\boldsymbol{w} = \{\boldsymbol{u}, \mathbf{V}, \boldsymbol{b}\}$. The NTK of this non-linear function is*

$$\langle \nabla_{\boldsymbol{w}} f_{\boldsymbol{w}}(\boldsymbol{x}), \nabla_{\boldsymbol{w}} f_{\boldsymbol{w}}(\boldsymbol{x}_t) \rangle = \sigma(\mathbf{V}\boldsymbol{x} + \boldsymbol{b})^\top \sigma(\mathbf{V}\boldsymbol{x}_t + \boldsymbol{b}) + \boldsymbol{u}^\top \boldsymbol{u}(\boldsymbol{x}^\top \boldsymbol{x}_t + 1)\sigma'(\mathbf{V}\boldsymbol{x} + \boldsymbol{b})^\top \sigma'(\mathbf{V}\boldsymbol{x}_t + \boldsymbol{b}),$$

*where $\langle \cdot, \cdot \rangle$ denotes Frobenius inner product or dot product depending on the context.*

The proof of this lemma can be found in Appendix B. Note that the encoder $\phi$ is no longer fixed, and we have $\phi_{\boldsymbol{\theta}}(\boldsymbol{x}) = \sigma(\mathbf{V}\boldsymbol{x} + \boldsymbol{b})$, where $\boldsymbol{\theta} = \{\mathbf{V}, \boldsymbol{b}\}$ are learnable parameters. By Lemma 3.1,

$$\langle \nabla_{\boldsymbol{w}} f_{\boldsymbol{w}}(\boldsymbol{x}), \nabla_{\boldsymbol{w}} f_{\boldsymbol{w}}(\boldsymbol{x}_t) \rangle = \phi_{\boldsymbol{\theta}}(\boldsymbol{x})^\top \phi_{\boldsymbol{\theta}}(\boldsymbol{x}_t) + \boldsymbol{u}^\top \boldsymbol{u}(\boldsymbol{x}^\top \boldsymbol{x}_t + 1)\phi_{\boldsymbol{\theta}}'(\boldsymbol{x})^\top \phi_{\boldsymbol{\theta}}'(\boldsymbol{x}_t). \tag{3}$$

Compared with the NTK in linear approximations (Equation (2)), Equation (3) has an additional term $\boldsymbol{u}^\top \boldsymbol{u}(\boldsymbol{x}^\top \boldsymbol{x}_t + 1)\phi_{\boldsymbol{\theta}}'(\boldsymbol{x})^\top \phi_{\boldsymbol{\theta}}'(\boldsymbol{x}_t)$, due to a learnable encoder $\phi_{\boldsymbol{\theta}}$. With sparse representations, we have $\phi_{\boldsymbol{\theta}}(\boldsymbol{x})^\top \phi_{\boldsymbol{\theta}}(\boldsymbol{x}_t) \approx 0$. However, it is not necessary true that $\boldsymbol{u}^\top \boldsymbol{u}(\boldsymbol{x}^\top \boldsymbol{x}_t + 1)\phi_{\boldsymbol{\theta}}'(\boldsymbol{x})^\top \phi_{\boldsymbol{\theta}}'(\boldsymbol{x}_t) \approx 0$ even when $\boldsymbol{x}$ and $\boldsymbol{x}_t$ are quite dissimilar, which violates Property 2.3.

To conclude, our analysis indicates that sparse representations alone are not effective enough to reduce forgetting in non-linear approximations.

## 4 OBTAINING SPARSITY WITH ELEPHANT ACTIVATION FUNCTIONS

Although Lemma 3.1 shows that the forgetting issue can not be fully addressed with sparse representations solely in deep learning methods, it also points out a possible solution: sparse gradients. With sparse gradients, we could have $\phi_{\boldsymbol{\theta}}'(\boldsymbol{x})^\top \phi_{\boldsymbol{\theta}}'(\boldsymbol{x}_t) \approx 0$. Together with sparse representations, we may still satisfy Property 2.3 in non-linear approximations, and thus reduce more forgetting. Specifically, we aim to design new activation functions to obtain both sparse representations and sparse gradients.

To begin with, we first define the sparsity of an function, which also applies to activation functions.

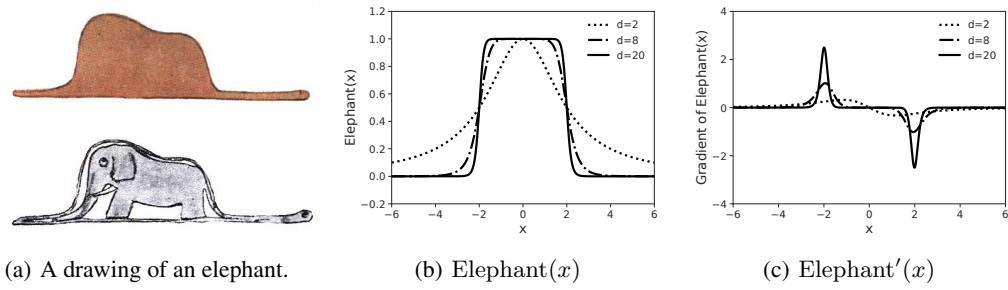

(a) A drawing of an elephant.      (b) Elephant$(x)$      (c) Elephant$'(x)$

Figure 1: (a) "My drawing was not a picture of a hat. It was a picture of a boa constrictor digesting an elephant." *The Little Prince*, by Antoine de Saint Exupéry. (b) Elephant functions with $a = 2$ and various $d$. (c) The gradient of elephant functions with $a = 2$ and various $d$.

**Definition 4.1** (sparse function). For a function $\sigma : \mathbb{R} \mapsto \mathbb{R}$, we define the sparsity of function $\sigma$ on input domain $[-C, C]$ as

$$S_{\epsilon, C}(\sigma) = \frac{|\{x \mid |\sigma(x)| \leq \epsilon, x \in [-C, C]\}|}{|\{x \mid x \in [-C, C]\}|} = \frac{|\{x \mid |\sigma(x)| \leq \epsilon, x \in [-C, C]\}|}{2C},$$

where $\epsilon$ is a small positive number and $C > 0$. As a special case, when $\epsilon \to 0^+$ and $C \to \infty$, define

$$S(\sigma) = \lim_{\epsilon \to 0^+} \lim_{C \to \infty} S_{\epsilon, C}(\sigma).$$

We call $\sigma$ a $S(\sigma)$-sparse function. In particular, $\sigma$ is called a sparse function if it is a 1-sparse function, i.e. $S(\sigma) = 1$.

**Remark 4.2.** Easy to verify that $0 \leq S(\sigma) \leq 1$. The sparsity of a function shows the fraction of nearly zero outputs given a symmetric input domain. For example, both $\text{ReLU}(x)$ and $\text{ReLU}'(x)$ are $\frac{1}{2}$-sparse functions. $\text{Tanh}(x)$ is a 0-sparse function while $\text{Tanh}'(x)$ is a sparse function. In the appendix, we summarize the results for common activation functions in Table 3 and visualize the activation functions with their gradients in Figure 5.

Next, we propose to use a novel class of bell-shaped activation functions, elephant activation functions [1]. Formally, an elephant function is defined as

$$\text{Elephant}(x) = \frac{1}{1 + \left| \frac{x}{a} \right|^d}, \tag{4}$$

where $a$ controls the width of the function and $d$ controls the slope. We call a neural network that uses elephant activation functions an *elephant neural network (ENN)*. For example, for MLPs and CNNs, we have *elephant MLPs (EMLPs)* and *elephant CNNs (ECNNs)*, respectively.

As shown in Figure 1, elephant activation functions have both sparse function values and sparse gradient values, which can also be formally proved.

**Lemma 4.3.** $\text{Elephant}(x)$ *and* $\text{Elephant}'(x)$ *are sparse functions.*

Specifically, $d$ controls the sparsity of gradients for elephant functions. The larger the value of $d$, the sharper the slope of an elephant function and the sparser the gradient. On the other hand, $a$ stands for the width of the function, controlling the sparsity of the function itself.

Next, we show that Property 2.3 holds under certain conditions for elephant functions.

**Theorem 4.4.** *Define $f_{\boldsymbol{w}}(\boldsymbol{x})$ as in Lemma 3.1. Let $\sigma$ be the elephant activation function with $d \to \infty$. When $|\mathbf{V}(\boldsymbol{x} - \boldsymbol{x}_t)| \succ 2a\mathbf{1}_m$, we have $\langle \nabla_{\boldsymbol{w}} f_{\boldsymbol{w}}(\boldsymbol{x}), \nabla_{\boldsymbol{w}} f_{\boldsymbol{w}}(\boldsymbol{x}_t) \rangle = 0$, where $\succ$ denotes an element-wise inequality symbol and $\mathbf{1}_m = [1, \cdots, 1]^\top \in \mathbb{R}^m$.*

**Remark 4.5.** Theorem 4.4 mainly proves that when $d \to \infty$, Property 2.3 holds for an EMLP with one hidden layer. However, even when $d$ is a small integer (e.g., 8), EMLPs still exhibit local elasticity, as we will show in the experiment section. The proof is in Appendix B.

---

[1] We name this bell-shaped activation function as the *elephant* function since the bell-shape is similar to the shape of an elephant (see Figure 1), honoring the work *The Little Prince* by Antoine de Saint Exupéry. This name also hints that this activation function empowers neural networks with continual learning ability, echoing the old saying that "an elephant never forgets".

Table 1: The test MSE of various networks in streaming learning for a simple regression task. Lower is better. All results are averaged over 5 runs, reported with standard errors.

| Method | Test Performance (MSE) |
|---|---|
| MLP (ReLU) | $0.4729 \pm 0.0110$ |
| MLP (Sigmoid) | $0.4583 \pm 0.0008$ |
| MLP (Tanh) | $0.4461 \pm 0.0013$ |
| MLP (ELU) | $0.4521 \pm 0.0019$ |
| SR-NN | $0.4061 \pm 0.0036$ |
| EMLP (Elephant) | $\mathbf{0.0081 \pm 0.0009}$ |

## 5 EXPERIMENTS

In this section, we first perform experiments in a simple regression task in the streaming learning setting. We will show that (1) sparse representations alone are not enough to address the forgetting issue, (2) ENNs can continually learn to solve regression tasks by reducing forgetting, and (3) ENNs are locally elastic even when $d$ is a small integer. Next, we show that by incorporating elephant functions, ENNs are able to achieve better performance than several baselines in class incremental learning, without utilizing pre-training, replay buffer, or task boundaries. Finally, we apply ENNs in reinforcement learning (RL) and demonstrate ENNs' advantage in reducing catastrophic forgetting.

### 5.1 STREAMING LEARNING FOR REGRESSION

In the real world, regression tasks are everywhere, from house price estimations (Madhuri et al. 2019) to stock predictions (Dase & Pawar 2010), weather predictions (Ren et al. 2021), and power consumption forecasts (Dmitri et al. 2016). However, most prior continual learning methods are designed for classification tasks rather than regression tasks, although catastrophic forgetting frequently arises in regression tasks as well (He & Sick 2021). In this section, we conduct experiments on a simple regression task in the streaming learning setting. In this setting, a learning agent is presented with one sample only at each time step and then performs learning updates given this new sample. Moreover, the learning process happens in a single pass of the whole dataset, that is, each sample only occurs once. Furthermore, the data stream is assumed to be non-iid. Finally, the evaluation happens after each new sample arrives, which requires the agent to learn quickly while avoiding catastrophic forgetting. Streaming learning methods enable real-time adaptation; thus they are more suitable in real-world scenarios where data is received in a continuous flow.

We consider streaming learning for approximating a sine function. In this task, there is a stream of data $(x_1, y_1), (x_2, y_2), \cdots, (x_n, y_n)$, where $0 \le x_1 < x_2 < \cdots < x_n \le 2$ and $y_i = \sin(\pi x_i)$. We set $n = 200$ in our experiment. The learning agent $f$ is an MLP with one hidden layer of size $1,000$. At each time step $t$, the learning agent only receives one sample $(x_t, y_t)$. We minimize the square error loss $l_t = (f(x_t) - y_t)^2$ with Adam optimizer (Kingma & Ba 2015), where $f(x_t)$ is the agent's prediction. During an evaluation, the agent performance is measured by the mean square error (MSE) on a test dataset with $1,000$ samples, where the inputs are evenly spaced over the interval $[0, 2]$.

We compare our method (EMLP) with two kinds of baselines. One is an MLP with classical activation functions, including ReLU, Sigmoid, ELU, and Tanh. The other is the sparse representation neural network (SR-NN) (Liu et al. 2019) which is designed to generate sparse representations. Specifically, we apply various classical activation functions (ReLU, Sigmoid, ELU, and Tanh) in SR-NNs. We set $d = 8$ for all elephant functions applied in EMLP.

We summarize test MSEs in Table 1. A lower MSE indicates better performance. Additional training details are included in the appendix. For the SR-NN, we present the best result among the combinations of SR-NNs and classical activation functions. Clearly, our method EMLP achieves the best performance, reaching a very low test MSE compared with baselines. Generally, we find that the SR-NN achieves slightly better performance than MLPs with classical activation functions, showing the benefits of sparse representations. However, compared with EMLP, the test MSE of the SR-NN is still large, indicating that the SR-NN fails to approximate $\sin(\pi x)$.

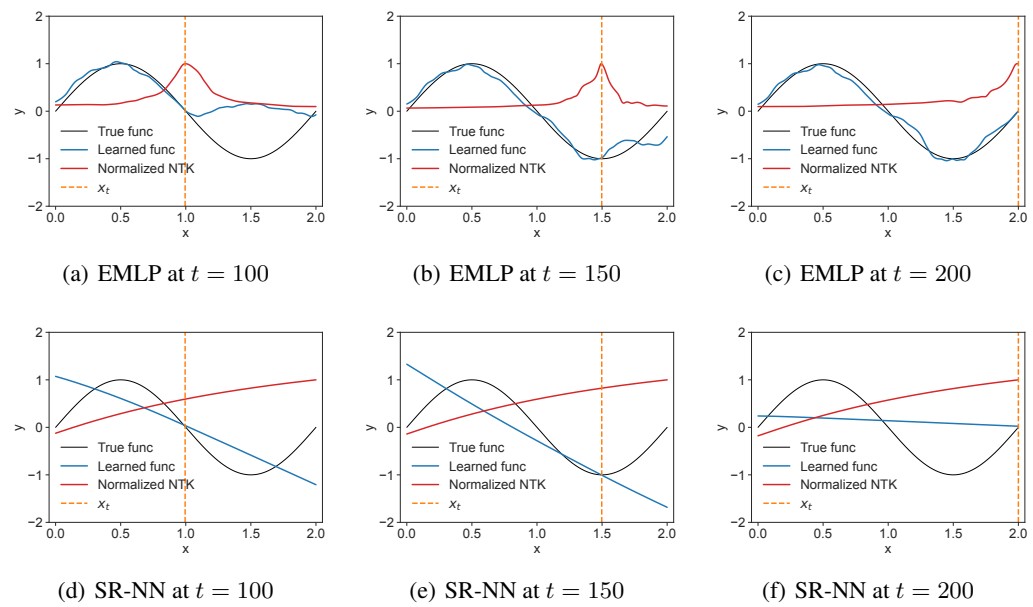

Figure 2: Plots of the true function $\sin(\pi x)$, the learned function $f(x)$, and the NTK function $\text{NTK}(x)$ at different training stages for EMLP and SR-NN. The NTK function $\text{NTK}(x)$ of the EMLP quickly reduces to 0 as $x$ moves away from $x_t$, demonstrating the local elasticity (Property 2.3) of EMLP.

Next, we plot the true function $\sin(\pi x)$, the learned function $f(x)$, and the NTK function $\text{NTK}(x) = \langle \nabla_{\boldsymbol{w}} f_{\boldsymbol{w}}(x), \nabla_{\boldsymbol{w}} f_{\boldsymbol{w}}(x_t) \rangle$ at different training stages for EMLP and SR-NN in Figure 2. The plots of MLP (ReLU), MLP (Sigmoid), MLP (Tanh), and MLP (ELU) are not presented, since they are similar to the plots of the SR-NN. The NTK function $\text{NTK}(\boldsymbol{x})$ is normalized such that the function value is in $[-1, 1]$. In Figure 2, the plots in the first row show that for EMLP with $d = 8$, $\text{NTK}(x)$ quickly decreases to 0 as $x$ moving away from $x_t$, demonstrating the local elasticity (Property 2.3) of EMLP with a small $d$. However, SR-NNs (and MLPs with classical activation functions) are not locally elastic; the learned function basically evolves as a linear function, a phenomenon often appears in over-parameterized neural networks (Jacot et al. 2018, Chizat et al. 2019).

*By injecting local elasticity to a neural network, we can break the inherent global generalization ability of the neural networks (Ghiassian et al. 2020), constraining the output changes of the neural network to small local areas. Utilizing this phenomenon, we can update a wrong prediction by "editing" outputs of a neural network nearly point-wisely.* To verify, we first train a neural network to approximate $\sin(\pi x)$ well in the traditional supervised learning style. We call the function the old learned function at this stage. Now assume that the original $y$ value of an input $x$ is changed to $y'$, while the true values of other inputs remain the same. The goal is to update the prediction of the neural network for input $x$ to this new value $y'$, while keeping the predictions of other inputs close to the original predictions, without expensive re-training on the whole dataset. Note that this requirement is common, especially in RL; and we will illustrate it with more details later. For now, we focus on supervised learning for a clearer demonstration. Specifically, we would like to update the output value at $x = 1.5$ of the learned function from $y = -1.0$ to $y' = -1.5$. We perform experiments on both EMLP and MLP, showing in Figure 3. Clearly, both neural works successfully update the prediction at $x = 1.5$ to the new value. However, besides the prediction at $x = 1.5$, the learned function of MLP is changed globally while the changes of EMLP are mainly confined in a small local area around $x = 1.5$. That is, we can successfully correct the wrong prediction nearly point-wisely by "editing" the output value for ENNs, but not for classical neural networks. To conclude, we summarize our findings in the following:

- Sparse representations are useful but are not effective enough to solve streaming regression tasks.
- EMLP is locally elastic even when $d$ is small (e.g., 8) and it allows "editing" the output value nearly point-wisely.
- EMLP can continually learn to solve regression tasks by reducing forgetting.

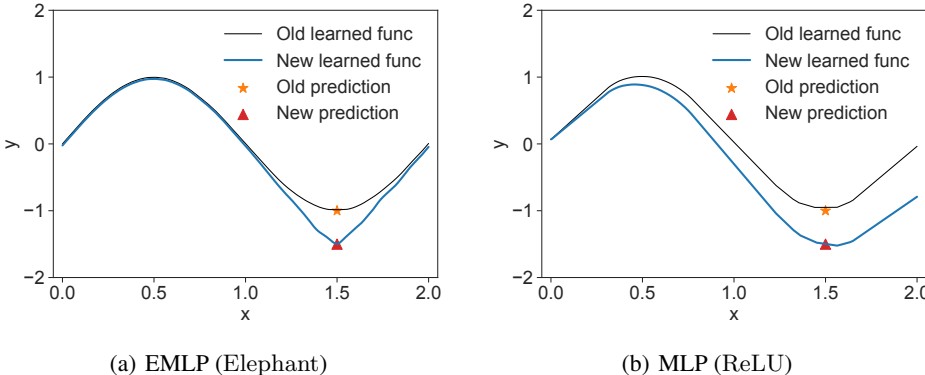

(a) EMLP (Elephant)  (b) MLP (ReLU)

Figure 3: A demonstration of local elasticity of the EMLP. EMLP allows updating the old prediction nearly point-wisely by "editing" the output value of the neural network. However, we cannot achieve this for MLP since it is not locally elastic.

## 5.2 CLASS INCREMENTAL LEARNING

In addition to regression tasks, our method can be applied to classification tasks as well, by simply replacing classical activation functions with elephant activation functions in a classification model. Though our model is agnostic to data distributions, we test it in class incremental learning in order to compare it with previous methods. Moreover, we adopt a stricter variation of the continual learning setting by adding the following restrictions: (1) same as streaming learning, each sample only occurs once during training, (2) task boundaries are not provided or inferred (Aljundi et al. 2019a), (3) neither pre-training nor a fixed feature encoder is allowed (Wolfe & Kyrillidis 2022), and (4) no buffer is allowed to store old task information in any form, such as training samples and gradients.

Surprisingly, we find no methods are designed for or have been tested in the above setting. As a variant of EWC (Kirkpatrick et al. 2017), Online EWC (Schwarz et al. 2018) almost meets these requirements, although it still requires task boundaries. To overcome this issue, we propose *Streaming EWC* as one of the baselines, which updates the fisher information matrix after every training sample. Streaming EWC can be viewed as a special case of Online EWC, treating each training sample as a new task. Besides Streaming EWC, we consider SDMLP (Bricken et al. 2023) and FlyModel (Shen et al. 2021) as two strong baselines, although they require either task boundary information or multiple data passes. Finally, two naive baselines are included, which train MLPs and CNNs without any techniques to reduce forgetting.

We test various methods on several standard datasets — Split MNIST (Deng 2012), Split CIFAR10 (Krizhevsky 2009), Split CIFAR100 (Krizhevsky 2009), and Split Tiny ImageNet (Le & Yang 2015). *We only present results on Split MNIST here due to page limit and put all other results in Appendix C.2.* For SDMLP and FlyModel, we take results from Bricken et al. (2023) directly. For MLP and CNN, we use ReLU by default. For our methods, we apply elephant functions with $d = 4$ in an MLP with one hidden layer and a simple CNN. The resulting neural networks are named EMLP and ECNN, respectively.

The test accuracy is used as the performance metric. A result summary of different methods on Split MNIST is shown in Table 2. The number of neurons refers to the size of the last hidden layer in a neural network, also known as the feature dimension. Overall, FlyModel performs the best, although it requires additional task boundary information. Our methods (EMLP and ECNN) are the second best without utilizing task boundary information by training for a single pass. Other findings are summarized in the following, reaffirming the findings in Mirzadeh et al. (2022a;b):

- Wider neural networks forget less by using more neurons.
- Neural network architectures can significantly impact learning performance: CNN (ECNN) is better than MLP (EMLP), especially with many neurons.
- Architectural improvement is larger than algorithmic improvement: using a better architecture (e.g., EMLP and ECNN) is more beneficial than incorporating Streaming EWC.

Overall, we conclude that applying elephant activation functions significantly reduces forgetting and boosts performance in class incremental learning under strict constraints.

Table 2: The test accuracy of various methods in class incremental learning on *Split MNIST*. Higher is better. All accuracies are averaged over 5 runs, reported with standard errors. The number of neurons refers to the size of the last hidden layer in a neural network, i.e. the feature dimension.

| Method | Neurons | Dataset Passes | Task Boundary | Test Accuracy |
|---|---|---|---|---|
| MLP | 1K | 1 | ✗ | 0.665±0.014 |
| MLP+Streaming EWC | 1K | 1 | ✗ | 0.708±0.008 |
| SDMLP | 1K | 500 | ✗ | 0.69 |
| FlyModel | 1K | 1 | ✓ | **0.77** |
| **EMLP (ours)** | 1K | 1 | ✗ | 0.723±0.006 |
| CNN | 1K | 1 | ✗ | 0.659±0.016 |
| CNN+Streaming EWC | 1K | 1 | ✗ | 0.716±0.024 |
| **ECNN (ours)** | 1K | 1 | ✗ | 0.732±0.007 |
| MLP | 10K | 1 | ✗ | 0.621±0.010 |
| MLP+Streaming EWC | 10K | 1 | ✗ | 0.609±0.013 |
| SDMLP | 10K | 500 | ✗ | 0.53 |
| FlyModel | 10K | 1 | ✓ | **0.91** |
| **EMLP (ours)** | 10K | 1 | ✗ | 0.802±0.002 |
| CNN | 10K | 1 | ✗ | 0.769±0.011 |
| CNN+Streaming EWC | 10K | 1 | ✗ | 0.780±0.010 |
| **ECNN (ours)** | 10K | 1 | ✗ | 0.850±0.004 |

## 5.3 REINFORCEMENT LEARNING

Recently, Lan et al. (2023) showed that the forgetting issue exists even in single RL tasks and it is largely masked by a large replay buffer. Without a replay buffer, a single RL task can be viewed as a series of related but different tasks without clear task boundaries (Dabney et al. 2021). For example, in temporal difference (TD) learning, the true value function is usually approximated by $v$ with bootstrapping: $v(S_t) \leftarrow R_{t+1} + \gamma v(S_{t+1})$, where $S_t$ and $S_{t+1}$ are two successive states and $R_{t+1} + \gamma v(S_{t+1})$ is named the TD target. During training, the TD target constantly changes due to bootstrapping, non-stationary state distribution, and changing policy. To speed up learning while reducing forgetting, it is crucial to update $v(S_t)$ to the new TD target without changing other state values too much (Lan et al. 2023), where local elasticity can help.

*By its very nature, solving RL tasks requires continual learning ability for both classification and regression.* Concretely, estimating value functions using TD learning is a non-stationary regression task; given a specific state, selecting the optimal action from a discrete action space is very similar to a classification task. In this section, we demonstrate that incorporating elephant activation functions helps reduce forgetting in RL under memory constraints. Specifically, we use deep Q-network (DQN) (Mnih et al. 2013; 2015) as an exemplar algorithm following Lan et al. (2023). Four classical RL tasks from Gym (Brockman et al. 2016) and PyGame Learning Environment (Tasfi 2016) are chosen: MountainCar-v0, Acrobot-v1, Catcher, and Pixelcopter. An MLP/EMLP with one hidden layer of size $1,000$ is used to parameterize the action-value function in DQN for all tasks. For all elephant functions used in EMLPs, $d = 4$. The standard buffer size is 1e4. For EMLP, we use a small replay buffer with size 32; for MLP, we consider two buffer sizes — 1e4 and 32. More details are included in the appendix.

The return curves of various methods are shown in Figure 4, averaged over 10 runs, where shaded areas represent standard errors and $m$ denotes the size of a replay buffer. Clearly, using a tiny replay buffer, EMLP ($m = 32$) outperforms MLP ($m = 32$) in all tasks. Moreover, except Pixelcopter, EMLP ($m = 32$) achieves similar performance compared with MLP ($m = 1e4$), although it uses a much smaller buffer. In summary, the experimental results further confirm the effectiveness of elephant activation functions in reducing catastrophic forgetting.

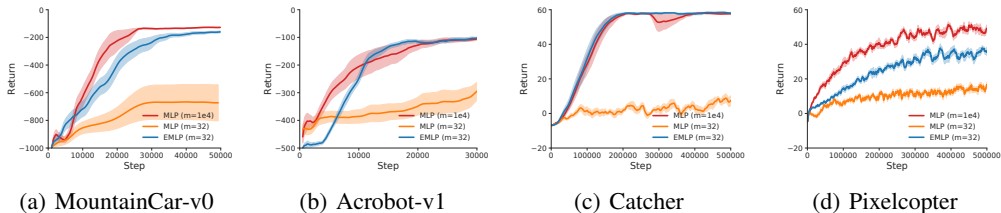

(a) MountainCar-v0       (b) Acrobot-v1       (c) Catcher       (d) Pixelcopter

Figure 4: The return curves of DQN in four tasks with different neural networks and buffer sizes. All results are averaged over 10 runs, where the shaded areas represent standard errors. Using a much smaller buffer, EMLP ($m = 32$) outperforms MLP ($m = 32$) and matches MLP ($m = 1e4$).

## 6 RELATED WORKS

**Architecture-based continual learning** Continual learning methods can be divided into several categories, such as regularization-based methods (Kirkpatrick et al. 2017, Schwarz et al. 2018, Zenke et al. 2017, Aljundi et al. 2019b), replay-based methods (Kemker et al. 2018, Farquhar & Gal 2018, Van de Ven & Tolias 2019, Delange et al. 2021), and optimization-based methods (Lopez-Paz & Ranzato 2017, Zeng et al. 2019, Farajtabar et al. 2020). We encourage readers to check recent surveys (Khetarpal et al. 2022, Delange et al. 2021, Wang et al. 2023) for more details. Here, we focus on architecture-based methods that are more related to our work. Among them, our work is inspired by Mirzadeh et al. (2022a;b), which study and analyze the effect of different neural architectures on continual learning. Several approaches involve careful selection and allocation of a subset of weights in a large network for each task (Mallya & Lazebnik 2018, Sokar et al. 2021, Fernando et al. 2017, Serra et al. 2018, Masana et al. 2021, Li et al. 2019, Yoon et al. 2018), or the allocation of a network that is specific to each task (Rusu et al. 2016, Aljundi et al. 2017). Some other methods expand a neural network dynamically (Yoon et al. 2018, Hung et al. 2019, Ostapenko et al. 2019). Finally, Shen et al. (2021) and Bricken et al. (2023) proposed novel neural network structures inspired by biological neural circuits, acting as two strong baseline methods in our experiments.

**Sparse representations** Sparse representations are known to help reduce forgetting for decades (French 1992). In supervised learning, sparse training (Dettmers & Zettlemoyer 2019, Liu et al. 2020, Sokar et al. 2021), dropout variants (Srivastava et al. 2013, Goodfellow et al. 2013, Mirzadeh et al. 2020, Abbasi et al. 2022, Sarfraz et al. 2023), and weight pruning methods (Guo et al. 2016, Frankle & Carbin 2019, Blalock et al. 2020, Zhou et al. 2020) are shown to speed up training and improve generalization. In continual learning, both SDMLP (Shen et al. 2021) and Fly-Model (Bricken et al. 2023) are designed to generate sparse representations. In RL, Le et al. (2017), Liu et al. (2019), and Pan et al. (2022) showed that sparse representations help stabilize training and improve overall performance.

**Local elasticity and memorization** He & Su (2020) proposed the concept of local elasticity. Chen et al. (2020) introduced label-aware neural tangent kernels, showing that models trained with these kernels are more locally elastic. Mehta et al. (2021) proved a theoretical connection between the scale of neural network initialization and local elasticity, demonstrating extreme memorization using large initialization scales. Incorporating Fourier features in the input of a neural network also induces local elasticity, which is greatly affected by the initial variance of the Fourier basis (Li & Pathak 2021).

## 7 CONCLUSION

In this work, we proposed elephant activation functions that can make neural networks more resilient to catastrophic forgetting. Specifically, we showed that incorporating elephant activation functions in classical neural networks helps improve learning performance in streaming learning for regression, class incremental learning, and reinforcement learning. Theoretically, our work provided a deeper understanding of the role of activation functions in catastrophic forgetting. We hope our work will inspire more researchers to study and design better neural network architectures for continual learning.

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

## A  FUNCTION SPARSITY AND GRADIENT SPARSITY OF VARIOUS ACTIVATION FUNCTIONS

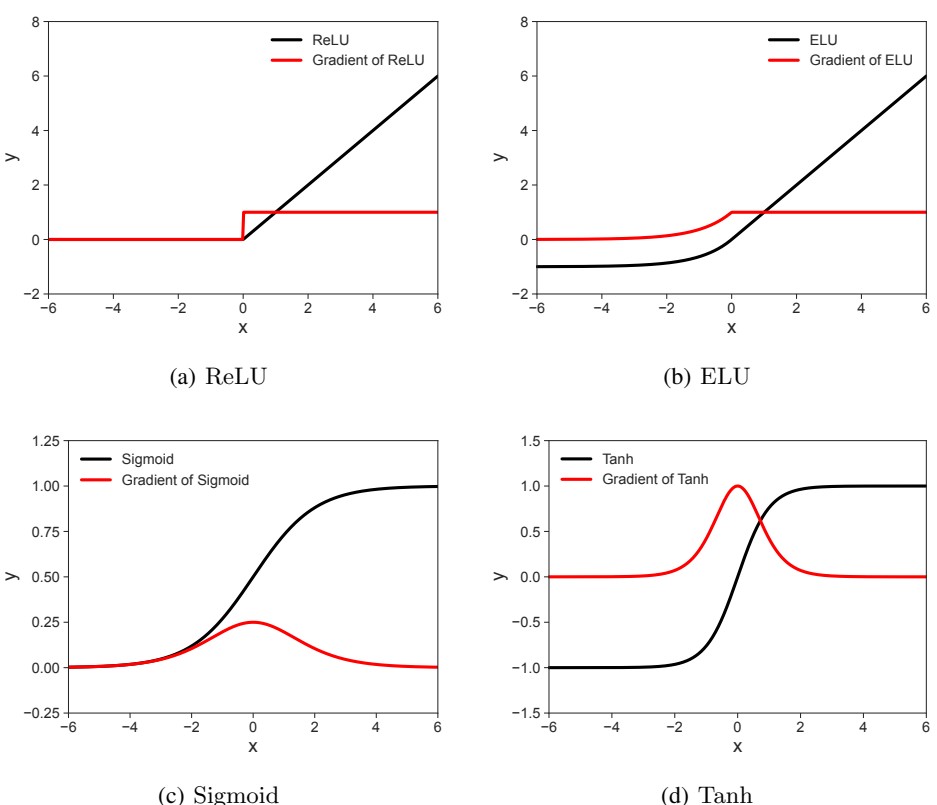

(a) ReLU

(b) ELU

(c) Sigmoid

(d) Tanh

Figure 5: Visualizations of common activation functions and their gradients.

Table 3: The function sparsity and gradient sparsity of various activation functions. Among them, only Elephant is sparse in terms of both function values and gradient values.

| activation | function sparsity | gradient sparsity |
|---|---|---|
| ReLU | 1/2 | 1/2 |
| Sigmoid | 1/2 | 1 |
| Tanh | 0 | 1 |
| ELU | 0 | 1/2 |
| Elephant | 1 | 1 |

## B  PROOFS

**Derivation of Equation (1)**   By Taylor expansion, we have

$$
\begin{aligned}
&f_{\boldsymbol{w}'}(\boldsymbol{x}) - f_{\boldsymbol{w}}(\boldsymbol{x}) \\
=&f_{\boldsymbol{w}+\Delta_{\boldsymbol{w}}}(\boldsymbol{x}) - f_{\boldsymbol{w}}(\boldsymbol{x}) \\
=&f_{\boldsymbol{w}}(\boldsymbol{x}) + \langle \nabla_{\boldsymbol{w}} f_{\boldsymbol{w}}(\boldsymbol{x}), \Delta_{\boldsymbol{w}} \rangle + O(\Delta_{\boldsymbol{w}}^2) - f_{\boldsymbol{w}}(\boldsymbol{x}) \\
=&\langle \nabla_{\boldsymbol{w}} f_{\boldsymbol{w}}(\boldsymbol{x}), \Delta_{\boldsymbol{w}} \rangle + O(\Delta_{\boldsymbol{w}}^2) \\
=&-\alpha \nabla_f L(f, F, \boldsymbol{x}_t) \langle \nabla_{\boldsymbol{w}} f_{\boldsymbol{w}}(\boldsymbol{x}), \nabla_{\boldsymbol{w}} f_{\boldsymbol{w}}(\boldsymbol{x}_t) \rangle + O(\Delta_{\boldsymbol{w}}^2).
\end{aligned}
$$

**Lemma 3.1.** *Given a non-linear function $f_{\boldsymbol{w}}(\boldsymbol{x}) = \boldsymbol{u}^\top \sigma(\mathbf{V}\boldsymbol{x} + \boldsymbol{b}) : \mathbb{R}^n \mapsto \mathbb{R}$, where $\sigma$ is a non-linear activation function, $\boldsymbol{x} \in \mathbb{R}^n$, $\boldsymbol{u} \in \mathbb{R}^m$, $\mathbf{V} \in \mathbb{R}^{m \times n}$, $\boldsymbol{b} \in \mathbb{R}^m$, and $\boldsymbol{w} = \{\boldsymbol{u}, \mathbf{V}, \boldsymbol{b}\}$. The NTK of this non-linear function is*

$$\langle \nabla_{\boldsymbol{w}} f_{\boldsymbol{w}}(\boldsymbol{x}), \nabla_{\boldsymbol{w}} f_{\boldsymbol{w}}(\boldsymbol{x}_t) \rangle = \sigma(\mathbf{V}\boldsymbol{x} + \boldsymbol{b})^\top \sigma(\mathbf{V}\boldsymbol{x}_t + \boldsymbol{b}) + \boldsymbol{u}^\top \boldsymbol{u}(\boldsymbol{x}^\top \boldsymbol{x}_t + 1)\sigma'(\mathbf{V}\boldsymbol{x} + \boldsymbol{b})^\top \sigma'(\mathbf{V}\boldsymbol{x}_t + \boldsymbol{b}),$$

*where $\langle \cdot, \cdot \rangle$ denotes Frobenius inner product or dot product given the context.*

*Proof.* Let $\circ$ denote Hadamard product. By definition, we have

$$\begin{aligned}
&\langle \nabla_{\boldsymbol{w}} f_{\boldsymbol{w}}(\boldsymbol{x}), \nabla_{\boldsymbol{w}} f_{\boldsymbol{w}}(\boldsymbol{x}_t) \rangle \\
&= \langle \nabla_{\boldsymbol{u}} f_{\boldsymbol{w}}(\boldsymbol{x}), \nabla_{\boldsymbol{u}} f_{\boldsymbol{w}}(\boldsymbol{x}_t) \rangle + \langle \nabla_V f_{\boldsymbol{w}}(\boldsymbol{x}), \nabla_V f_{\boldsymbol{w}}(\boldsymbol{x}_t) \rangle + \langle \nabla_b f_{\boldsymbol{w}}(\boldsymbol{x}), \nabla_b f_{\boldsymbol{w}}(\boldsymbol{x}_t) \rangle \\
&= \langle \sigma(\mathbf{V}\boldsymbol{x} + \boldsymbol{b}), \sigma(\mathbf{V}\boldsymbol{x}_t + \boldsymbol{b}) \rangle + \langle \boldsymbol{u} \circ \sigma'(\mathbf{V}\boldsymbol{x} + \boldsymbol{b})\boldsymbol{x}^\top, \boldsymbol{u} \circ \sigma'(\mathbf{V}\boldsymbol{x}_t + \boldsymbol{b})\boldsymbol{x}_t^\top \rangle \\
&\quad + \langle \boldsymbol{u} \circ \sigma'(\mathbf{V}\boldsymbol{x} + \boldsymbol{b}), \boldsymbol{u} \circ \sigma'(\mathbf{V}\boldsymbol{x}_t + \boldsymbol{b}) \rangle \\
&= \sigma(\mathbf{V}\boldsymbol{x} + \boldsymbol{b})^\top \sigma(\mathbf{V}\boldsymbol{x}_t + \boldsymbol{b}) + (\boldsymbol{x}^\top \boldsymbol{x}_t + 1) (\boldsymbol{u} \circ \sigma'(\mathbf{V}\boldsymbol{x} + \boldsymbol{b}))^\top (\boldsymbol{u} \circ \sigma'(\mathbf{V}\boldsymbol{x}_t + \boldsymbol{b})), \\
&= \sigma(\mathbf{V}\boldsymbol{x} + \boldsymbol{b})^\top \sigma(\mathbf{V}\boldsymbol{x}_t + \boldsymbol{b}) + \boldsymbol{u}^\top \boldsymbol{u}(\boldsymbol{x}^\top \boldsymbol{x}_t + 1)\sigma'(\mathbf{V}\boldsymbol{x} + \boldsymbol{b})^\top \sigma'(\mathbf{V}\boldsymbol{x}_t + \boldsymbol{b}).
\end{aligned}$$

$\square$

**Lemma 4.3.** Elephant$(x)$ *and* Elephant$'(x)$ *are sparse functions.*

*Proof.* $\text{Elephant}(x) = \frac{1}{1 + \left|\frac{x}{a}\right|^d}$ and $\left|\text{Elephant}'(x)\right| = \frac{d}{a}\left|\frac{x}{a}\right|^{d-1}\left(\frac{1}{1 + \left|\frac{x}{a}\right|^d}\right)^2$. For $0 < \epsilon < 1$, easy to verify that

$$|x| \geq a(\frac{1}{\epsilon} - 1)^{1/d} \implies \text{Elephant}(x) \leq \epsilon \quad \text{and} \quad |x| \geq \frac{d}{2\epsilon} \implies \left|\text{Elephant}'(x)\right| \leq \epsilon.$$

For $C > a(\frac{1}{\epsilon} - 1)^{1/d}$, we have $S_{\epsilon,C}(\text{Elephant}) \geq \frac{C - a(\frac{1}{\epsilon} - 1)^{1/d}}{C}$, thus

$$S(\text{Elephant}) = \lim_{\epsilon \to 0^+} \lim_{C \to \infty} S_{\epsilon,C}(\text{Elephant}) \geq \lim_{\epsilon \to 0^+} \lim_{C \to \infty} \frac{C - a(\frac{1}{\epsilon} - 1)^{1/d}}{C} = \lim_{\epsilon \to 0^+} 1 = 1.$$

Similarly, for $C > \frac{d}{2\epsilon}$, we have $S_{\epsilon,C}(\text{Elephant}') \geq \frac{C - \frac{d}{2\epsilon}}{C}$, thus

$$S(\text{Elephant}') = \lim_{\epsilon \to 0^+} \lim_{C \to \infty} S_{\epsilon,C}(\text{Elephant}') \geq \lim_{\epsilon \to 0^+} \lim_{C \to \infty} \frac{C - \frac{d}{2\epsilon}}{C} = \lim_{\epsilon \to 0^+} 1 = 1.$$

Note that $S(\text{Elephant}) \leq 1$ and $S(\text{Elephant}') \leq 1$. Together, we conclude that $S(\text{Elephant}) = 1$ and $S(\text{Elephant}') = 1$; Elephant$(x)$ and Elephant$'(x)$ are sparse functions. $\square$

**Theorem 4.4.** *Define $f_{\boldsymbol{w}}(\boldsymbol{x})$ as it in Lemma 3.1. Let $\sigma$ be the elephant activation function with $d \to \infty$. When $|\mathbf{V}(\boldsymbol{x} - \boldsymbol{x}_t)| \succ 2a\mathbf{1}_m$, we have $\langle \nabla_{\boldsymbol{w}} f_{\boldsymbol{w}}(\boldsymbol{x}), \nabla_{\boldsymbol{w}} f_{\boldsymbol{w}}(\boldsymbol{x}_t) \rangle = 0$, where $\succ$ denotes an element-wise inequality symbol and $\mathbf{1}_m = [1, \cdots, 1] \in \mathbb{R}^m$.*

*Proof.* When $d \to \infty$, the elephant function is a rectangular function, i.e.

$$\sigma(x) = \text{rect}(x) = \begin{cases} 1, & |x| < a, \\ \frac{1}{2}, & |x| = a, \\ 0, & |x| > a. \end{cases}$$

In this case, it is easy to verify that $\forall x, y \in \mathbb{R}$, $|x - y| > 2w$, we have $\sigma(x)\sigma(y) = 0$ and $\sigma'(x)\sigma'(y) = 0$. Denote $\Delta_{\boldsymbol{x}} = \boldsymbol{x} - \boldsymbol{x}_t$. Then when $|\mathbf{V}\Delta_{\boldsymbol{x}}| \succ 2a\mathbf{1}_m$, we have $\sigma(\mathbf{V}\boldsymbol{x} + \boldsymbol{b})^\top \sigma(\mathbf{V}\boldsymbol{x}_t + \boldsymbol{b}) = 0$ and $\sigma'(\mathbf{V}\boldsymbol{x} + \boldsymbol{b})^\top \sigma'(\mathbf{V}\boldsymbol{x}_t + \boldsymbol{b}) = 0$. In other words, when $\boldsymbol{x}$ and $\boldsymbol{x}_t$ are dissimilar in the sense that $|\mathbf{V}(\boldsymbol{x} - \boldsymbol{x}_t)| \succ 2a\mathbf{1}_m$, we have $\langle \nabla_{\boldsymbol{w}} f_{\boldsymbol{w}}(\boldsymbol{x}), \nabla_{\boldsymbol{w}} f_{\boldsymbol{w}}(\boldsymbol{x}_t) \rangle = 0$.

$\square$

## C    EXPERIMENTAL DETAILS AND ADDITIONAL RESULTS

The source code will be released upon acceptance. All our experiments are conducted on CPUs or V100 GPUs. The computation to repeat all experiments in this work is not high (much less 0.1 GPU years). However, the exact number of used computation is hard to estimate.

For all ENNs, we only replace the activation functions of the last hidden layer with elephant activation functions. [2] Before training, the weight values in all layers are initialized from $U(-\sqrt{k}, \sqrt{k})$, where $k = 1/\text{in\_features}$. For the bias values in the layer where elephant functions are used, we initialize them with evenly spaced numbers over the interval $[-\sqrt{3}\sigma_{bias}, \sqrt{3}\sigma_{bias}]$, where $\sigma_{bias}$ is the standard deviation of the bias values. The bias values are initialized in such way so that ENNs can generate diverse features. All other bias values are initialized with 0s.

As a limitation, there is a lack of a theoretical way to set $\sigma_{bias}$ or $a$ in elephant activation functions appropriately. The best values seem to be depend on the input data distribution as well as the number of the input features (i.e., in\_features). In practice, we choose $\sigma_{bias}$ and $a$ from a small set.

To improve sample efficiency, we update a model for multiple times given each new sample/mini-batch data. We call this number the update epoch $E$.

### C.1    STREAMING LEARNING FOR REGRESSION

We list the (swept) hyper-parameters in streaming learning for regression in Table 4. Among them, $d$, $a$, and $\sigma_{bias}$ are specific hyper-parameters for ENNs, where $\lambda$ and $\beta$ are two hyper-parameters used in SR-NN Liu et al. (2019).

Table 4: The (swept) hyper-parameters in streaming learning for regression.

| Hyper-parameter | Value |
|---|---|
| Optimizer | Adam |
| learning rate | $\{3e-3, 1e-3, 3e-4, 1e-4, 3e-5, 1e-5\}$ |
| $E$ | 10 |
| $d$ | 8 |
| $a$ | $\{0.02, 0.04, 0.08, 0.16, 0.32\}$ |
| $\sigma_{bias}$ | $\{0.08, 0.16, 0.32, 0.64, 1.28\}$ |
| Set KL loss weight $\lambda$ | $\{0, 0.1, 0.01, 0.001\}$ |
| SR-NN $\beta$ | $\{0.05, 0.1, 0.2\}$ |

### C.2    CLASS INCREMENTAL LEARNING

#### C.2.1    EXPERIMENTAL RESULTS ON SPLIT MNIST AND SPLIT CIFAR10

For both MNIST and CIFAR10, we split them into 5 sub-datasets, and each of them contains two of the classes. The CNN model consists of a convolution layer, a max pooling operator, and a linear layer. We also try CNNs with more convolution layers but find the performance is worse and worse as we increase the number of convolution layers. Besides simple CNNs, we test ConvMixer (Trockman & Kolter 2022) which can achieve around $92.5\%$ accuracy in just 25 epochs in classical supervised learning [3]. However, we find that ConvMixer completely fails in class incremental learning setting, as shown in Table 5 and Table 6. Same as Mirzadeh et al. (2022b), the results show that using more advanced models does not necessarily lead to better performance in class incremental learning.

In Figure 6, we plot the test accuracy curves during training on Split MNIST and Split CIFAR10 when the number of neurons is $10K$. All results are averaged over 5 runs and the shaded regions represent standard errors.

---

[2]The initial experiments with EMLPs on Split MNIST show that replacing all activation functions with elephant functions hurts performance, probably due to loss of plasticity. We leave this for future work.

[3]https://github.com/locuslab/convmixer-cifar10

Table 5: The test accuracy of various methods in class incremental learning on *Split MNIST*. Higher is better. All accuracies are averaged over 5 runs, reported with standard errors.

| Method | Neurons | Dataset Passes | Task Boundary | Test Accuracy |
|---|---|---|---|---|
| MLP | 1K | 1 | ✗ | 0.665±0.014 |
| MLP+Streaming EWC | 1K | 1 | ✗ | 0.708±0.008 |
| SDMLP | 1K | 500 | ✗ | 0.69 |
| FlyModel | 1K | 1 | ✓ | **0.77** |
| **EMLP (ours)** | 1K | 1 | ✗ | 0.723±0.006 |
| CNN | 1K | 1 | ✗ | 0.659±0.016 |
| CNN+Streaming EWC | 1K | 1 | ✗ | 0.716±0.024 |
| **ECNN (ours)** | 1K | 1 | ✗ | 0.732±0.007 |
| ConvMixer | 1K | 1 | ✗ | 0.110±0.003 |
| EConvMixer (ours) | 1K | 1 | ✗ | 0.105±0.003 |
| MLP | 10K | 1 | ✗ | 0.621±0.010 |
| MLP+Streaming EWC | 10K | 1 | ✗ | 0.609±0.013 |
| SDMLP | 10K | 500 | ✗ | 0.53 |
| FlyModel | 10K | 1 | ✓ | **0.91** |
| **EMLP (ours)** | 10K | 1 | ✗ | 0.802±0.002 |
| CNN | 10K | 1 | ✗ | 0.769±0.011 |
| CNN+Streaming EWC | 10K | 1 | ✗ | 0.780±0.010 |
| **ECNN (ours)** | 10K | 1 | ✗ | 0.850±0.004 |
| ConvMixer | 10K | 1 | ✗ | 0.107±0.003 |
| EConvMixer (ours) | 10K | 1 | ✗ | 0.104±0.003 |

Table 6: The test accuracy of various methods in class incremental learning on *Split CIFAR10*, averaged over 5 runs. Higher is better. Standard errors are also reported.

| Method | Neurons | Test Accuracy |
|---|---|---|
| MLP | 1K | 0.151±0.008 |
| MLP+Streaming EWC | 1K | 0.158±0.005 |
| **EMLP (ours)** | 1K | **0.197±0.003** |
| CNN | 1K | 0.151±0.001 |
| CNN+Streaming EWC | 1K | 0.147±0.010 |
| **ECNN (ours)** | 1K | **0.192±0.004** |
| ConvMixer | 1K | 0.100±0.0027 |
| EConvMixer (ours) | 1K | 0.100±0.0001 |
| MLP | 10K | 0.173±0.004 |
| MLP+Streaming EWC | 10K | 0.169±0.003 |
| **EMLP (ours)** | 10K | **0.239±0.002** |
| CNN | 10K | 0.151±0.001 |
| CNN+Streaming EWC | 10K | 0.179±0.007 |
| **ECNN (ours)** | 10K | **0.243±0.002** |
| ConvMixer | 10K | 0.102±0.0015 |
| EConvMixer (ours) | 10K | 0.100±0.0001 |

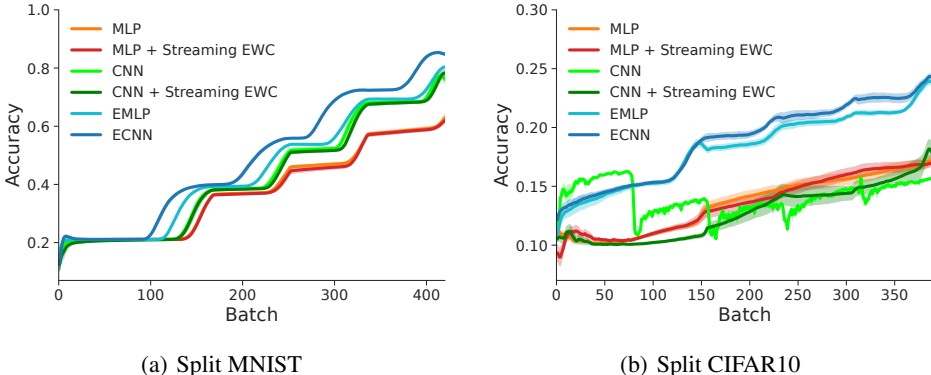

(a) Split MNIST          (b) Split CIFAR10

Figure 6: The test accuracy curves during training. The x-axis shows the number of training mini-batches. All results are averaged over 5 runs and the shaded regions represent standard errors.

Table 7: The test accuracy of various methods in class incremental learning on *Split Embedding CIFAR10*, averaged over 5 runs. Higher is better. Standard errors are also reported.

| Method | Neurons | Dataset Passes | Task Boundary | Test Accuracy |
|---|---|---|---|---|
| MLP | 1K | 1 | ✗ | 0.662±0.006 |
| SDMLP | 1K | 2000 | ✗ | 0.56 |
| FlyModel | 1K | 1 | ✓ | 0.69 |
| **EMLP (ours)** | 1K | 1 | ✗ | **0.726±0.008** |
| MLP | 10K | 1 | ✗ | 0.697±0.002 |
| SDMLP | 10K | 2000 | ✗ | 0.77 |
| FlyModel | 10K | 1 | ✓ | **0.82** |
| **EMLP (ours)** | 10K | 1 | ✗ | 0.755±0.002 |

Table 8: The test accuracy of various methods in class incremental learning on *Split Embedding CIFAR100*, averaged over 5 runs. Higher is better. Standard errors are also reported.

| Method | Neurons | Dataset Passes | Task Boundary | Test Accuracy |
|---|---|---|---|---|
| MLP | 1K | 1 | ✗ | 0.157±0.001 |
| SDMLP | 1K | 500 | ✗ | 0.39 |
| FlyModel | 1K | 1 | ✓ | 0.36 |
| **EMLP (ours)** | 1K | 1 | ✗ | **0.391±0.003** |
| MLP | 10K | 1 | ✗ | 0.425±0.001 |
| SDMLP | 10K | 500 | ✗ | 0.43 |
| FlyModel | 10K | 1 | ✓ | **0.58** |
| **EMLP (ours)** | 10K | 1 | ✗ | 0.449±0.002 |

Table 9: The test accuracy of various methods in class incremental learning on *Split Embedding Tiny ImageNet*, averaged over 5 runs. Higher is better. Standard errors are also reported.

| Method | Neurons | Dataset Passes | Task Boundary | Test Accuracy |
|---|---|---|---|---|
| MLP | 10K | 1 | ✗ | **0.249±0.003** |
| **EMLP (ours)** | 10K | 1 | ✗ | 0.241±0.002 |
| MLP | 100K | 1 | ✗ | 0.264±0.002 |
| **EMLP (ours)** | 100K | 1 | ✗ | **0.350±0.002** |

### C.2.2 COMBINING WITH PRE-TRAINING

To show that our method can be combined with pre-training, we relax our experimental assumptions by adding the pre-training technique.

To be specific, for CIFAR10 and CIFAR100, we use 256 dimensional latent embeddings which are provided by Bricken et al. (2023), taken from the last layer of a frozen ConvMixer (Trockman & Kolter 2022) that is pre-trained on ImageNet32 (Chrabaszcz et al. 2017). The corresponding embedding datasets are called *Embedding CIFAR10* and *Embedding CIFAR100*, respectively. We split Embedding CIFAR10 into 5 sub-datasets while Embedding CIFAR100 is split into 50 sub-datasets; each sub-dataset contains two of the classes. We test different methods on the two datasets and present results in Table 7 and Table 8. Note that the the results of SDMLP and FlyModel are from Bricken et al. (2023).

For Tiny ImageNet, we use 768 dimensional latent embeddings which are taken from the last layer of a frozen ConvMixer (Trockman & Kolter 2022) that is pre-trained on ImageNet-1k (Russakovsky et al. 2015), provided by Hugging Face [4]. The corresponding embedding dataset is called *Embedding Tiny ImageNet*. We then split Embedding Tiny ImageNet into 100 sub-datasets; each of them contains two of the classes. The experimental results are presented in Table 9.

Overall, the performance of EMLP is greatly improved with the help of pre-training. For example, the test accuracy on CIFAR10 is boosted from 25% to 75%. Moreover, EMLP still outperforms MLP significantly, showing that our method can be combined with common practices in class incremental learning, such as pre-training.

### C.2.3 HYPER-PARAMETERS

We list the (swept) hyper-parameters for Split MNIST and Split CIFAR10 in Table 10. Among them, $d$, $a$, and $\sigma_{bias}$ are specific hyper-parameters for ENNs, where $\gamma$ and $\lambda$ are two hyper-parameters used in (streaming) EWC (Kirkpatrick et al. 2017).

Table 10: The (swept) hyper-parameters for *Split MNIST* and *Split CIFAR10*.

| Hyper-parameter | Value |
|---|---|
| Optimizer | RMSProp with decay=0.999 |
| learning rate | $\{3e-6, 1e-6, 3e-7, 1e-7, 3e-8, 1e-8\}$ |
| mini-batch size | 125 |
| $E$ | $\{1, 2\}$ |
| $d$ | 4 |
| $a$ | $\{0.02, 0.04, 0.08, 0.16, 0.32\}$ |
| $\sigma_{bias}$ | $\{0.04, 0.08, 0.16, 0.32, 0.64\}$ |
| EWC $\gamma$ | $\{0.5, 0.8, 0.9, 0.95, 0.99, 0.999\}$ |
| EWC $\lambda$ | $\{1e1, 1e2, 1e3, 1e4, 1e5\}$ |

Similarly, the (swept) hyper-parameters for Split Embedding CIFAR10, Split Embedding CIFAR100, and Split Embedding Tiny ImageNet are listed in Table 11 and Table 12.

---

[4] https://huggingface.co/timm/convmixer_768_32.in1k

Table 11: The (swept) hyper-parameters for *Split Embedding CIFAR10* and *Split Embedding CIFAR100*.

| Hyper-parameter | Value |
|---|---|
| Optimizer | RMSProp with decay=0.999 |
| learning rate | $\{1e-4, 3e-5, 1e-5, 3e-6, 1e-6, 3e-7, 1e-7\}$ |
| mini-batch size | 125 |
| $E$ | $\{1, 2, 4\}$ |
| $d$ | 4 |
| $a$ | $\{0.02, 0.04, 0.08, 0.16, 0.32\}$ |
| $\sigma_{bias}$ | $\{0.04, 0.08, 0.16, 0.32, 0.64\}$ |

Table 12: The (swept) hyper-parameters for *Split Embedding Tiny ImageNet*.

| Hyper-parameter | Value |
|---|---|
| Optimizer | RMSProp with decay=0.999 |
| learning rate | $\{1e-5, 3e-6, 1e-6, 3e-7, 1e-7\}$ |
| mini-batch size | 250 |
| $E$ | $\{2, 4, 8\}$ |
| $d$ | 4 |
| $a$ | $\{0.08, 0.16, 0.32, 0.64\}$ |
| $\sigma_{bias}$ | $\{0.32, 0.64, 1.28, 2.56\}$ |

## C.3 REINFORCEMENT LEARNING

We list the (swept) hyper-parameters for reinforcement learning in Table 13. Among them, $d$, $a$, and $\sigma_{bias}$ are specific hyper-parameters for ENNs. Other hyper-parameters and training settings can be found in Lan et al. (2020).

Table 13: The (swept) hyper-parameters in reinforcement learning.

| Hyper-parameter | Value |
|---|---|
| Optimizer | RMSProp with decay=0.999 |
| learning rate | $\{1e-2, 3e-3, 1e-3, 3e-4, 1e-4, 3e-5, 1e-5, 3e-6\}$ |
| mini-batch size | 32 |
| $E$ | $\{1, 2\}$ |
| $d$ | 4 |
| $a$ | $\{0.08, 0.16, 0.32, 0.64\}$ |
| $\sigma_{bias}$ | $\{0.08, 0.16, 0.32, 0.64, 1.28\}$ |
| discount factor $\gamma$ | 0.99 |

