# OpenReview forum: "Elephant Neural Networks: Born to Be a Continual Learner"
_ICLR.cc/2024/Conference — Submitted to ICLR 2024_

### Official Review · Reviewer_rwQQ · 2023-10-30

**Soundness:** 3 good
**Presentation:** 3 good
**Contribution:** 2 fair
**Rating:** 5
**Confidence:** 4

**Summary:**

After showing that sparsity of the activation function and its derivative are desired for alleviation of catastrophic forgetting, the paper proposes a symmetric activation function for neural networks with hyper-parameters for adjusting the degree of neuron plasticity.  Experimental evaluation shows that while the proposed new models do not necessarily beat the existing baseline method, they come very close, while providing the advantage of not requiring explicit signalling to the model that it's learning a new task.

**Strengths:**

Ability to control generalisation vs. plasticity through hyper-parameter of an activation function is a great feature for continual learning, though it comes at a cost of more hyper-parameters.

The sparsity analysis of activation functions is interesting.

The proposed method does not require task boundary information, which is very desired for practical continual learning.

Empirical evaluation is honest - it shows that proposed methods does not beat (on accuracy) other baselines, though those other baselines come at a price of task boundary information requirement, which the proposed method does not incur.

**Weaknesses:**

Switch to a symmetric activation function will have significant impact on the internal representation (and perhaps generalisation capabilities) of a neural network.  After all, there must be a reason why we typically use asymmetric activation functions like ReLU, sigmoid, tanh.  I think any proposal of a radically new (and here I mean the symmetry aspect) activation function calls for additional evaluations of the utility of the proposed activation function under normal (all data available) conditions, just to convince the reader that we are not loosing much by using the new activation function.

Also, feels like increase of sparsity of the gradient will have severe effect on learning dynamics in very deep networks - after all, there is a reason why we use ReLU and not sigmoid.  Perhaps trade off between depth (can I still adjust early layers) and forgetting (can I remember previous training samples) is inevitable.

This is not a significant issue, but while the "Little Prince" reference is cute, I don't find the name "elephant" for the proposed activation function appropriate - the shape of that function looks nothing like Exupery's elephant inside a boa... and I think the point of the function is the sparsity of its gradient, not the shape of the function itself...which still doesn't has nothing to do with an elephant.

**Questions:**

It feels like the propose activation turns the neurons into RBF like units, kind of Gaussian-like kernel, though not quite, since it measures decaying distance to points only along 1D.   While I like the fact that we can control the width and steepness of the elephant function for the fine tuning of network's plasticity, it seems that one fixed value of d for the entire network might be not appropriate.  Perhaps different tuning at different layers?  Also, it seems that for continual learning setting different d at different time of training might be appropriate.  Any thoughts on this?

Remark 4.2 should the  "that" be "if"?

Concept of 1-sparse and 1/2 sparse...I am not quite getting it - perhaps a bit more explanation is in order.

Can these "elephant" activation networks, in principle (i.e. not in a continual learning scenario), achieve same generalisation as equivalent ReLU networks?

---

### Official Review · Reviewer_nrBP · 2023-10-31

**Soundness:** 2 fair
**Presentation:** 2 fair
**Contribution:** 2 fair
**Rating:** 1
**Confidence:** 5

**Summary:**

This paper studies the role of activation function as a means to mitigate catastrophic forgetting. It finds that both sparse representations and gradients play critical roles in mitigating forgetting. It proposes a new activation function called elephant activation that promotes sparsity for both representations and gradients, making neural networks more resilient to catastrophic forgetting. Neural networks with elephant activation function named as elephant neural networks (ENNs) demonstrate efficacy in various learning scenarios e.g., regression, class incremental learning and reinforcement learning.

**Strengths:**

This paper studies the interaction between neural network architectures and continual learning which is under-studied. It focuses on activation function - a critical component of neural networks. It is a good problem and sparsity has been called out in some of the earliest continual learning papers in the deep learning era as a good inductive bias for continual learning, e.g., Kemker et al. (AAAI 2018).

The paper presents theoretical analysis to demonstrate that sparse representations alone cannot mitigate forgetting in non-linear approximations. They discover that both sparse representations and gradients are essential to mitigate forgetting in continual learning.

It introduces a new class of activation function to enforce sparsity for both representations and gradients to mitigate catastrophic forgetting. And, it provides formal proof that both elephant activation and its gradients are sparse functions.

Various networks with elephant activation (EMLP, ECNN) demonstrate better performance than their counterparts with classical activations as well as sparse neural networks (sparse representation only) in regression, class incremental learning and reinforcement learning tasks.

**Weaknesses:**

In non-rehearsal methods, elephant neural networks (ENNs) demonstrate efficacy. However, it is unknown how ENNs generalize to other performant algorithms e.g., rehearsal based methods. It is unclear if ENNs’ efficacy is specific to the particular non-rehearsal settings (Sec. 5.2)  or not!

Besides MLP and CNN, efficacy of elephant activation with ViTs is unexplored and necessary to verify its generality.

Lack of experiments on high dimensional and large scale datasets e.g., ImageNet-1K. Also it is unclear how ENNs perform in other data orderings e.g., IID (classes revisit) besides class-IL which is an extreme edge case. An ideal continual learner should be robust to any data orderings including class-IL and IID. Continual learning is about knowledge accumulation, and it conflates the test with an adversarial distribution with the goal.

It is unclear if elephant activation increases computational and/or memory overheads or not compared to commonly used activation functions in terms of learning efficiency (updates).

A common practice in measuring catastrophic forgetting is to compare continual learner with an offline learner (jointly trained on all data) given that they both use the same architecture. Therefore, a more precise difference between continual learner and offline learner (i.e., forgetting) can be made when they both use elephant neural networks e.g., EMLPs or ECNNs. Currently, it is missing in the paper as offline learner’s performance is not mentioned.

The way the main text is written leads to an understanding that elephant activations are used in all layers replacing classical activations. However, surprisingly, it is written in supplemental material (Sec. C) that in all cases, elephant activations are only used in the last hidden layer whereas all other layers have classical activations. It is mentioned in the footnote that replacing all activations with elephant activations hurts performance. This highly undermines this paper’s core contributions and efficacy of the proposed method. These limitations should be mentioned and highlighted in the main paper. Otherwise readers will be highly misled if they read the main text.

It is unclear whether the performance gain originates from elephant activations or not. More experimental verification is required to establish that elephant activation mainly reduces forgetting. It is also unclear how sufficient sparsity to tackle forgetting is enforced with limited use of elephant activations.

**Questions:**

Why does elephant activation only work in the last hidden layer and remain ineffective in other layers?

Why is it not mentioned in the main paper that elephant activation in conjunction with regular activations work effectively otherwise it fails?

How does the use of elephant activation in the last layer only mitigate catastrophic forgetting? Is it an artifact of a particular experimental setup?

---

> ### Comment · Reviewer_nrBP · 2023-11-22
>
> The authors have not provided a rebuttal, but I did review the comments from the other reviewers. My rating is unchanged.

---

### Official Review · Reviewer_HrPu · 2023-11-04

**Soundness:** 2 fair
**Presentation:** 2 fair
**Contribution:** 2 fair
**Rating:** 3
**Confidence:** 5

**Summary:**

In this work, authors introduce a sparse activation function known as elephant. The proposed activation function is tested on streaming data such as sequential regression, classification and RL to show the proposed activation better results compared to conventional approaches.

**Strengths:**

1. Introduces sparse activation function

**Weaknesses:**

1. Experiment setup and text is slightly misleading, theoretically it is not possible that the activation function alone can resolve forgetting, activation patterns, weights should be parameterized to avoid neural cross talk across talk. The paper would benefit from showing results on deeper architecture and SoTA continual learning approaches.
2. Writing should be improved, keywords such as excellent should be avoided given results are almost 13% lower than SoTA.
3. Ablation study missing
4. Comparsion missing against sparse activation functions such as K-Winner take all, L-WTA etc.

**Questions:**

There is a large body of theoretical work neglected in this work. As the issue of catastrophic forgetting is well studied theoretically [5-12], so I am not sure why authors claims forgetting in NNs is not well understood.

Second using sparsity to overcome forgetting is well studied [1-3].

Third [1] have shown better performance on split-mnist with replay buffer, only one pass and without any task boundaries, so I am not sure one can claim they have excellent work, given current performance reported in this work is <80%.

To better understand Forgetting one should report forgetting ratio, backward transfer, and forward transfer (for instance, look at gradient episodic memory paper). Can authors report their task matrix? Currently I am not sure what benefit elephant activation offers beside sparsity that would potentially overcome forgetting.

Empirical evaluation of activation function is already done by goodfellow in 2013, so it is not new that sparsity alone cannot solve forgetting. One should also compare model’s with k-winner take all activation function, since that also adds sparsity. Can author report how elephant performs compared to K-WTA?
It would have been interesting to show how your model works with SoTA continual learning systems such as Gdumb, A-GEM, iCARL etc, current experiments are limited to shallow model and thus it is unclear how would model work with deeper structure.

What is the level of sparsity achieved by the network using proposed activation function? Can authors provide details about it?

Finally, an ablation study should be conducted with various hyper-parameters of ENN. It is still not clear how stable the proposed activation is.



1.	Ororbia, A., Mali, A., Giles, C.L. and Kifer, D., 2022. Lifelong neural predictive coding: Learning cumulatively online without forgetting. Advances in Neural Information Processing Systems, 35, pp.5867-5881.
2.	Sokar, G., Mocanu, D.C. and Pechenizkiy, M., 2022, September. Avoiding Forgetting and Allowing Forward Transfer in Continual Learning via Sparse Networks. In Joint European Conference on Machine Learning and Knowledge Discovery in Databases (pp. 85-101). Cham: Springer Nature Switzerland.
3.	Von Oswald, J., Zhao, D., Kobayashi, S., Schug, S., Caccia, M., Zucchet, N. and Sacramento, J., 2021. Learning where to learn: Gradient sparsity in meta and continual learning. Advances in Neural Information Processing Systems, 34, pp.5250-5263.
4.	Schwarz, J., Jayakumar, S., Pascanu, R., Latham, P.E. and Teh, Y., 2021. Powerpropagation: A sparsity inducing weight reparameterisation. Advances in neural information processing systems, 34, pp.28889-28903.
5.	Doan, T., Bennani, M.A., Mazoure, B., Rabusseau, G. and Alquier, P., 2021, March. A theoretical analysis of catastrophic forgetting through the ntk overlap matrix. In International Conference on Artificial Intelligence and Statistics (pp. 1072-1080). PMLR.

6.	Raghavan, K. and Balaprakash, P., 2021. Formalizing the generalization-forgetting trade-off in continual learning. Advances in Neural Information Processing Systems, 34, pp.17284-17297.

7.	Evron, I., Moroshko, E., Ward, R., Srebro, N. and Soudry, D., 2022, June. How catastrophic can catastrophic forgetting be in linear regression?. In Conference on Learning Theory (pp. 4028-4079). PMLR.

8.	Mirzadeh, S.I., Chaudhry, A., Yin, D., Hu, H., Pascanu, R., Gorur, D. and Farajtabar, M., 2022, June. Wide neural networks forget less catastrophically. In International Conference on Machine Learning (pp. 15699-15717). PMLR.

9.	Braun, L., Dominé, C., Fitzgerald, J. and Saxe, A., 2022. Exact learning dynamics of deep linear networks with prior knowledge. Advances in Neural Information Processing Systems, 35, pp.6615-6629.

10.	Lin, S., Ju, P., Liang, Y. and Shroff, N., 2023. Theory on Forgetting and Generalization of Continual Learning. arXiv preprint arXiv:2302.05836.

11.	Andle, J. and Yasaei Sekeh, S., 2022, October. Theoretical understanding of the information flow on continual learning performance. In European Conference on Computer Vision (pp. 86-101). Cham: Springer Nature Switzerland.

12.	Heckel, R., 2022, May. Provable continual learning via sketched Jacobian approximations. In International Conference on Artificial Intelligence and Statistics (pp. 10448-10470). PMLR.

---

> ### Comment · Reviewer_HrPu · 2023-11-22
> **Discussion**
>
> Hi All,
>
> I have read other reviews, and authors haven't submitted any response, thus I maintain my rating.
>
> Thanks

---

### Meta-Review · Area_Chair_dvuG · 2023-12-06

**Metareview:**

The paper introduces an intruiging approach for continual learning based on a novel activation function. Reviewers have pointed out significant concerns that were not addressed during the discussion phase. As such I have to recommend rejection at this stage.

**Justification For Why Not Higher Score:**

Lack of rebuttal for important issues.

**Justification For Why Not Lower Score:**

N/A

---

### Decision · Program_Chairs · 2024-01-16

Reject